# *Streptococcus mutans* Associated with Endo-Periodontal Lesions in Intact Teeth

Alessio Buonavoglia [1,†] , Adriana Trotta [2,*,†] , Michele Camero [2] , Marco Cordisco [2] , Michela Maria Dimuccio [2] and Marialaura Corrente [2]

1   Department of Biomedical and Neuromotor Sciences, School of Dentistry, 40125 Bologna, Italy
2   Department of Veterinary Medicine, University of Bari "Aldo Moro", Str. Prov. per Casamassima, Km 3, 70010 Valenzano, Italy
*   Correspondence: adriana.trotta@uniba.it or trotta.adri@gmail.com; Tel.: +39-33-3682-8764
†   These authors contributed equally to this work.

**Abstract:** A massive periodontal destruction can affect the root canal (RC) system and potentially expose the pulpo-dentinal complex to opportunistic bacteria. *Streptococcus mutans* is a major pathogen of human caries and periodontal diseases, and its virulence mostly resides in the ability to adhere to collagen and form biofilms, due to collagen-binding proteins (CBPs) Cnm and Cbm. Seventeen patients affected by severe endo-periodontal lesions without caries and/or exposure of pulpal tissue were subjected to tooth extraction and samples for microbiological investigation were collected from the root surface (RS) and RC. The collected swabs were cultivated and subjected to the quantitative real-time PCR (qPCR) for the detection of *S. mutans* and to the PCR for the *cnm/cbm* genes investigation, followed by next-generation sequencing (NGS). *S. mutans* DNA was detected in 12/17 (70.5%) RS samples and in 8/17 (47.0%) RC samples. In the CBPs screening of positive samples, the *cnm* gene was detected in 4/12 (33.3%) RS and in 1/8 (12.5%) RC samples, whilst all the samples tested negative for the *cbm* gene. The presence of the *cnm* gene could enhance the local virulence of the pathogens. Therefore, *S. mutans* have to be included as potential periodontopathogen bacterium in severe or refractory forms of periodontal diseases.

**Keywords:** *streptococcus mutans*; collagen-binding proteins; endo-periodontal infection

## 1. Introduction

The oral cavity houses the second largest and most complex microbial communities after the gut, harboring about $10^{10}$ microorganisms [1]. It hosts a huge variety of microorganisms, which include bacteria, fungi, viruses, and protozoa. The mouth, which has a large composition of ecology, is a sophisticated ecosystem where microbes could colonize the hard surfaces of the teeth and the soft tissues of the oral mucosa [1,2]. Moreover, the resident bacterial species cannot arrive to the pulpal tissues thanks to the external hard tissues and may become pathogenic only after they breach the barrier of the commensals, causing infection and disease [3]. Therefore, the pulpal tissue of healthy subjects is sterile. However, in pathological situations, which can affect the enamel and/or the cementum, the pulpo-dentinal complex could be exposed to the bacterial aggression, and it could be placed at risk of colonization and infection [3]. Caries are the most common cause of pulp exposure and subsequent endodontic bacterial infection, resulting from a complex interaction of specific host immunity, bad oral care and hygiene, diet, and resident microbial composition [1,3]. Moreover, a massive periodontal destruction can secondarily affect the root canal (RC), forming a pathologic communication between pulpal and periodontal tissues with the proliferation of bacteria, which can result in pulpal inflammation and necrosis. Periodontal and endodontic tissues have strict anatomical correlation through lateral and accessory canals, apical foramen, and dentinal tubules, where bacteria may migrate from one tissue to the other [4]. Over local manifestations and loss of tooth-periodontium

integrity, periodontal and endodontic infections may evolve into potential life-threatening infections such as abscesses or disseminating infections with systemic involvement, especially in some categories of patients (immunocompromised subjects, with high risk of infective endocarditis, patients with organic dysfunctions) [5,6].

*Streptococcus mutans* is generally considered to be a major pathogen of human caries [7] and is also involved in periodontal diseases [8]. *S. mutans* strains are classified into serotypes c, e, f, and k on the basis of the biochemical composition of serotype-specific polysaccharides. From the literature, it is known that approximately 70–80% of strains found in the human oral cavity are classified as serotype c, followed by the serotype e (~20%), and the serotypes f and k (less than 5% each) [7].

The virulence of *S. mutans* as a dental pathogen mostly resides in its ability to (i) adhere and form biofilms on tooth surfaces, (ii) produce large quantities of organic acids, and (iii) tolerate low pH and oxidative stress. The oral host tissue attachment could be due to a group of glycotransferases, which act as a cell surface protein antigen, recognized to have a collagen-binding activity [9]. The most described collagen-binding proteins (CBPs), described only in the *S. mutans* strains, are the Cnm and Cbm proteins [10]. These act as adhesins, i.e., they have the ability to adhere to type I collagen, thus promoting the biofilm formation, the development, and the duration of the periodontal pathologies [11]. In addition, *S. mutans* strains can became invasive, and the *cnm* gene is required for *S. mutans* adherence to endothelial and epithelial cells and intracellular invasion, also being an important virulence factor for systemic disease secondary to bacteremia such as endocarditis [10–12]. The distribution frequency of the strains carrying the *cnm* gene among oral isolates has been estimated to be approximately 10–20% and the detection of *S. mutans* in the oral sites has been subject of interest, not only due to its primary role in caries onset but also due to its association with extra-oral infections. In particular, the serotype k was the most recent serotype, and it was recognized to possess the most prominent being as the defect of the glucose side chain in serotype-specific rhamnose–glucose polymers, which is related to the highest incidence of this serotype in the cardiovascular specimens [7]. These findings suggest that serotype k *S. mutans* possibly has a high level of virulence for systemic diseases. In the present study, the presence of *S. mutans*, as the only bacterial pathogenic species, in teeth extracted from patients with endodontic-periodontal lesions (EPL), without caries and/or exposure of pulpal tissue, was investigated by means of traditional culture-method and biomolecular tests in order to establish the association between the *S. mutans* and the EPL. In addition, the presence of CBP-positive strains was investigated.

## 2. Materials and Methods

### 2.1. Patients' Information

From December 2021 to July 2022, n = 17 patients aged from 20 to 65 years who were affected by severe EPL were enrolled in this study. The exclusion criteria for this study were having received antibiotic therapy up to 3 months before clinical examination, systemic diseases, and pregnancy. Other exclusion criteria were the presence of caries, pulpal exposure, old restorations, and/or root damage (fracture, cracking, perforations, or external resorptions) affecting the tooth with EPL. Study participants were healthy patients categorized under American society of Anesthesiologists (ASA 1) and involved in the research signing a formal written informed consent form. During the clinical examination, the most represented signs and symptoms were spontaneous pain or pain on palpation/percussion, purulent exudate/suppuration, tooth mobility, sinus tract, crown and/or gingival color alterations, and bone resorption in the apical or furcation region.

Periodontal probing revealed deep (>6 mm) periodontal pockets with a decreased attachment level >5 mm in >1 tooth surface. Pulpal necrosis was diagnosed with non-responsivity to thermal and electric pulp sensibility tests. On intraoral inspection and radiographic evaluation, teeth do not present decay or coronal leakages. According to the classification of periodontal and peri-implant diseases and conditions of the American Academy of Periodontology [4], the final diagnosis for every subject was endo-periodontal

lesion Grade 3 without root damage, with a primary involvement from periodontal tissues and pulpal necrosis.

Among the 17 patients, the 17 teeth affected were: 4 lower incisors, 2 lower canines, 3 lower first molars, 2 upper second premolars, 2 upper first molars, 2 upper second molars, and 2 upper third molars. According to the patients, extraction was planned because the severe periodontal involvement could have hindered any periodontal or endodontic treatment. After extraction, teeth were examined to exclude decays or coronal leakages (Figure 1).

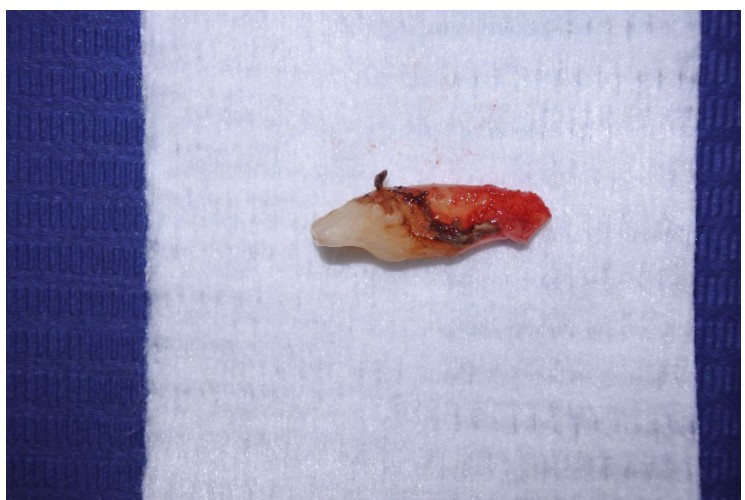

**Figure 1.** Extracted tooth without appreciable decay or leakages on crown surface.

### 2.2. Sampling Collection

Sampling collection for traditional culture-based methods and biomolecular investigations was performed immediately after extraction. Firstly, a sterile swab was used to scrub on RS, especially in root zones where biofilms were visible, and subsequently collected in sterile tubes with transport medium Tryptic Soy Agar (TSA) (Nuova Aptaca, Canelli, Italy). The samples were stored at −20 °C until further processed. Subsequently, the crown was disinfected with 2.5% NaOCl (Niclor 2.5, Ogna, Muggiò, Italy) for 30 s. Preparation of the access cavity was carried out using a sterile high-speed diamond bur under sterile saline solution. Before the pulp chamber was exposed, the cleaning of the tooth was repeated as previously described. Pulpal tissue observed was clinically evaluable as non-bleeding, confirming pulpal necrosis. After gently irrigation with sterile saline solution, a sterile #10 K-type stainless hand file (Maillefer, Ballaigues, Switzerland) was introduced into the canal to the level of the tooth apex. The working length was estimated with preoperative radiography. In multirooted teeth, samples were collected from the largest RC. After a gentle scraping along the RC walls to collect bacteria in the medium, paper points #15 (Dentsply-Maillefer, Ballaigues, Switzerland) were positioned as deep as possible in the canal for 60 s, and then they were collected in sterile tubes and subsequently stored at −20 °C until further analyses.

### 2.3. Bacteriological Analysis

The analyses were performed in the laboratory of bacteriology and biomolecular analysis of the Department of Veterinary Medicine (DVM), University of Bari (Italy). The swabs were collected before the biomolecular investigation the samples from RS, and paper points from RC were subjected to a culture-based method with a protocol useful to identify the main pathogenic bacterial species of the oral microflora. All the samples were plated on 5% Columbia blood agar (CBA), McConkey agar (MCK), Mannitol salt agar (MSA), and Tryptic Soy broth (TSB) (Liofilchem, Teramo, Italy), incubated in aerobic condition at 37 °C for 24–48 h. In addition, the CBA plates were also incubated into anaerobic and

microaerobic condition at 37 °C for 24–48 h. The initial presumptive identification of streptococci suspected colonies was performed using the macro-method tests (oxidase and catalase testes), gram staining, and biochemical macro-method tests (Api 20 Strep systems) purchased by Liofilchem (Teramo, Italy).

### 2.4. Biomolecular Analyses for the Detection of S. mutans DNA

The nucleic acid from 17 swabs of RS and 17 paper points of RC was isolated using the Power Soil Kit (Qiagen, Milan Italy). The protocol used was purchased by the supplier and the presence of *S. mutans* DNA in the specimens was determined by qPCR using the *S. mutans*-specific primers and probe as described by Yoshida and co-authors (2003) [13], with some modifications. Briefly, the detection of bacterial DNA was investigated using the primer set Smut3368-Forwar (5′-GCCTACAGCTCAGAGATGCTATTCT-3′) and Smut3481-Reverse (5′-GCCATACACCACTCATGAATTGA-3′) with the fluorescent probe Smut3423-T (5′-FAM-TGGAAATGACGGTCGCCGTTATGAA-TAMRA-3′), targeting the specific *gtfB* gene. Specifically, for each real-time PCR, 20 µL of a mixture containing 1 µL of extracted DNA, 1 µL TaqMan Universal PCR Master Mix (Applied Biosystems), 500 nM (each) sense and antisense primer, and 2500 nM TaqMan probe was placed in each well of a 96-well plate. Amplification and detection were performed using the C100 Thouch Thermal Cycler (Biorad, Segrate, Italy) with the following cycle profile: 95 °C for 10 min, and 45 cycles of 95 °C for 15 s and 58 °C for 1 min. As positive control the DNA of ATCC-700610 *S. mutans* strains as provided by LGC, Italy, was used, whilst sterile DNA-asi/RNA-asi free water (Qiagen, Italy) was used as negative control. The estimated amplicon size product was 114 bp.

### 2.5. Detection of S. mutans cnm and cbm Genes

In the specimens, among the samples which tested positive for the *S. mutans* DNA, the presence of the genes *cnm* and *cbm* was investigated by using a specific PCR with the set of primers as described by Nomura and co-authors [9]. Briefly, the detection of *cbm* was done with *cbm-Forward* (5′-GAT GGT ACC TAT GTTGAT TTG-3′) and *cbm-Reverse* (5′-CCG GTA ACG TTA TGG AGA TTA TTG-3′) sets of primer targeting the specific *cbm* gene, and the detection of *cnm* was performed using *cnm-CF* (forward) (5′-CTG AGG TTA CTG TCGTTA AA-3′) and *cnm-CR* (reverse) (5′-CAC TGT CTA CAT AAGCAT TC-3′) set of primers targeting the specific *cnm* gene. Estimated amplification size of the PCR amplified products for *cbm*-specific and *cnm*-specific sets of primers were 393 and 579 bp, respectively. PCR amplification was performed in a total volume of 25 µL composed of 23 µL of mixture plus 2 µL of template solution using the TaKaRa Ex Taq (Takara Bio, Jesi, Italy). The PCR amplification reaction was performed with the following cycling parameters: an initial denaturation at 95 °C for 2 min and then 35 cycles consisting of 94 °C for 30 s, 60 °C for 30 s, and 68 °C for 1 min, with a final extension at 68 °C for 7 min. The PCR amplified products were subjected to gel electrophoresis in a 1.5% agarose gel-Tris-acetate EDTA buffer. The gel was stained with 0.5 lg ethidium bromide per mL and photographed under UV illumination. A 100-bp DNA ladder (Biorad, Italy) was used as the molecular size standard. The sensitivity of the PCR assay was determined by using *cnm*-positive and *cbm*-positive strains from laboratory collection. The amplified products of PCR were purified by using enzymes QIA-quick PCR Purification Kit (Qiagen, Germantown, TN, USA), and specifically, the amplicons with the correct size were purified using the specific enzymes: Exonuclease 1 (Exo1) and Fast Alkaline Phosphatase (FastAP) purchased by a Qiaquick PCR Purification Kit (Qiagen GmbH, Hilden, Germany), which let us remove primer dimers (dNTPs) and/or aspecific bands (incorrect size). Briefly, PCR enzymes purification was performed in a total volume of 15 µL composed of 5 µL of mixture (0.5 µL of Exo1, 0.5 µL of Buffer, 1 µL of FastAP, and 3 µL of sterile RNAase and DNAase free water), plus 10 µL of the template solution, and the PCR amplification reaction was performed with the following cycling parameters: 20 min at 37 °C and 15 min at 85 °C. The purified PCR products with sufficient DNA concentrations (>10 ng/µL) were subsequently subjected

to sequence analysis, which was performed using the MiSeq NGS (Illumina, San Diego, CA, USA) technology. All sequences were analyzed with the Geneious Prime version 2022.1.1 Software and compared with reference sequences available on the BLAST database (https://blast.ncbi.nlm.nih.gov/Blast.cgi, accessed on 10 October 2022).

## 3. Results

The presence of alive *S. mutans* strains was observed by the culture-based method and among all the tested samples n = 34. The growth of *S. mutans* isolates in CBA incubated in aerobic and anaerobic condition was confirmed in 10 samples, in which the pure colonies were optimally identified by biochemical tests. In particular, *S. mutans* was isolated from 4 RC samples (patients n = 2; n = 6; n = 7; n = 16) and from 6 RS samples (patients n = 4; n = 6; n = 7; n = 10; n = 14; n = 15). In particular, a positive result was found in 12 out of 17 (70.5%) of patient's samples from RS and in 8 out of 17 (47.0%) samples from RC. *S. mutans* DNA was detected from all the RC systems belonging to teeth that were also positive for *S. mutans* DNA on RS. Moreover, 4 patients resulted positive for *S. mutans* DNA only in the samples collected from RS. The thermal cycle (CT) medium value of the samples was 35, with a minimum of 25 and a maximum of 40.

*Results of S. mutans cnm and cbm Genes*

Regarding the *cbm* and *cnm* genes investigation, 4 out of 12 (33.3%) *S. mutans* positive samples from RS were positive for the presence of the *cnm* gene and 1 out of 8 (12.5%) *S. mutans* positive samples from RC were positive for the *cnm* gene. The RC sample (n = 1) *cnm*-positive belonging to the tooth *cnm*-positive was also in the RS. All the samples tested negative for the *cbm* gene. Sequence analysis of the *cnm* gene PCR products revealed the complete matching of the amplicons with that of the *cnm* gene of *S. mutans* published in the databases. Table 1 summarizes all the results of traditional and biomolecular methods.

**Table 1.** *Streptococcus mutans* and CBP$_S$ genes detection by cultivation method and qPCR and PCR, respectively, in teeth extracted from patients affected by EPL.

| | **Patients** | | | | | | | | | | | | | | | | |
|---|---|---|---|---|---|---|---|---|---|---|---|---|---|---|---|---|---|
| | **1** | **2** | **3** | **4** | **5** | **6** | **7** | **8** | **9** | **10** | **11** | **12** | **13** | **14** | **15** | **16** | **17** |
| Tooth * | 33 | 41 | 16 | 32 | 27 | 15 | 28 | 28 | 46 | 36 | 31 | 17 | 33 | 42 | 25 | 26 | 36 |
| *S. mutans* Cultivation and Biomolecular Investigation | | | | | | | | | | | | | | | | | |
| *S.mutans* | | | | | | | | | | | | | | | | | |
| RC | − | ++ | − | + | − | ++ | ++ | − | + | − | + | − | + | − | − | ++ | − |
| RS | − | + | − | ++ | + | ++ | ++ | − | + | ++ | + | − | + | ++ | ++ | + | − |
| *S. mutans* CBPs Genes Investigation | | | | | | | | | | | | | | | | | |
| *cnm* gene | | | | | | | | | | | | | | | | | |
| RC | − | − | − | − | − | − | − | − | − | − | + | − | − | − | − | − | − |
| RS | − | − | − | − | + | − | − | − | + | − | + | − | + | − | − | − | − |
| *cbm* gene | | | | | | | | | | | | | | | | | |
| RC | − | − | − | − | − | − | − | − | − | − | − | − | − | − | − | − | − |
| RS | − | − | − | − | − | − | − | − | − | − | − | − | − | − | − | − | − |

* Type of tooth affected by EPL using FDI World Dental Federation notation. In the section "*S. mutans* isolation and biomolecular investigation": + = positive for the *S. mutans* qPCR/no growth observed in traditional culture. ++ = positive for the *S. mutans* qPCR/growth of *S. mutans* isolate observed in traditional media and confirmed with biochemical identification. − = negative for the *S. mutans* qPCR/no growth observed in traditional culture. In the section "*S. mutans* CBPs genes investigation": + = positive for the *S. mutans* CBP$_S$ genes detection by specific PCR targeting the *cnm* and *cbm* genes, respectively. − = negative for the *S. mutans* CBP$_S$ genes detection. RC: root canals; RS: root surface.

## 4. Discussion

Periodontal disease determines the exposure of anatomical communications between periodontal tissues and RC system. Alveolar bone resorption causes exposure and loss of cementum that exposes the dentinal tubules and allows bacterial entrance into the tooth [14].

Moreover, in severe periodontal disease where the depth of the periodontal pocket reach root apex, there is a direct communication between RC system and periodontal pocket with damage of neurovascular bundles and bacterial invasion from the pocket through the apex in the RC system. Over vascular supply, lateral and accessory canals may distribute bacteria and toxins from the periodontal apparatus into the dental pulp [15]. The formation of pathologic communication between the pulpal and periodontal tissues determines an EPL that can be classified according to periodontal probing, presence/absence of root damage and presence/absence of periodontitis. EPL can also be classified in three grades of severity. Grade 1 EPL presents a narrow deep (>6 mm) periodontal pocket that affects 1 tooth surface. In Grade 2 there is a wide, deep periodontal pocket in 1 tooth surface. In Grade 3, deep periodontal pockets are present in >1 tooth surface [4]. Primary involvement can originate from endodontic or periodontal tissues or with a combined origin. Pulpal pathology of periodontal origins is more controversial, because generally, negative caries teeth can present coronal leakages that make a clear diagnosis and etiopatho-genesis difficult. As observed in this research, all teeth selected in the study were negative for caries, pulpal exposures, root damage, and old restorations that could justify pulpal necrosis. It is suggested that a severe periodontal lesion can secondarily involve endodontic tissues. In this study, the presence of *S. mutans* strains has been investigated by using traditional culture-based methods and biomolecular methods. In the past, the study of oral microbiome was limited to the sole use of conventional culture-dependent techniques, but in some cases the abundant oral microflora could not be cultured for different reasons. The discovery of new genomic technologies, including NGS and bioinformatics, has finally revealed the complexities of the oral microbiome, thus providing a powerful means of studying the whole microbial communities [2]. In addition, next-generation techniques could help to understand the different composition of the oral microbiome in health subjects and in injured patients, which will give further information to explore the functional and metabolic alterations associated with the pathological cases and to identify molecular signatures for antibiotics and drug development, as well as to specifically targeting the therapies [1]. In this study, the use of NGS targeting the highly conserved specie-specific *gtfB* gene target and sequence technologies have been chosen for their sensitivity and specificity compared to traditional culture-based approaches, which still remains fundamental to confirm the presence of the alive bacteria. In particular, although a few growth media are available for selective isolation of the cariogenic Streptococci, it is still unclear as to which is the most sensile and efficacious [16]. Moreover, each species requires a specific protocol, and this greatly lengthens the time required for diagnosis. Therefore, the use of targeting-specific PCR, when applicable, could be useful to precisely detect the microbial species, especially for detection of specific bacterial genes, in a shorter time period. In this case, we were able to isolate, with a culture-based method, the *S. mutans* strains only in 50% of samples in which the *S. mutans* DNA was observed, therefore the two different methods must be considered as complementary tests and when possible, they have to be both carried out in order to obtain a result as specific/sensitive as possible in a shorter possible time. Moreover, *S. mutans* can migrate from periodontal pocket into endodontic system without clinically coronal leakages such as caries, traumatic tooth fractures, or old restorations. It is well-known that *S. mutans cnm*-positive strains have the ability to invade the endothelial cells with a subsequent periodontal vascular damage, and the possibility to migrate in bloodstreams with potential life-threatening pathologies like infective endocarditis or disseminating infections, especially in patients with pre-existing cardiovascular conditions or compromised immune system [11,17]. The expression of *cnm* gene in *S. mutans* enhances its local virulence with capability of epithelial and endothelial invasion, and this pathogen has to be considered as a potential periodonto-pathogen in severe or refractory forms of periodontal diseases. Of note, *S. mutans cnm*-positive isolate found on RS samples of patient n = 4 was found to express a different genetic pattern compared to the one detected in the endodontic micro-environment. Interestingly, only in 1 sample was the positivity to the *cnm* gene contemporarily detected in RC and RS samples of the same patient. We hypothesize

that the acquired properties obtained from this gene detection have less utility in necrotic endodontic tissues with lack of pulpal bloodstream. Instead, the function of the *cbm* gene is not yet well-defined, but it is thought that it also guarantees adhesiveness, invasiveness in endothelial and epithelial cells, and virulence to *S. mutans* strains.

The presence of *S. mutans* DNA was detected in most RS-analyzed samples of patients affected by severe periodontal disease. It is critical to recognize the interrelationship of endodontic and periodontal disease for successful management of these lesions. In fact, a non-treated endodontic infection may cause a delay or failure in periodontal healing, even in cases where the periodontal treatment has already been initiated, for the reinfection of periodontal space. On the other hand, a hidden periodontal lesion can reinfect a treated root canal system, invalidating endodontic treatment. This is not always observed in clinical practice, probably due to the fact that not all bacteria strains seem to possess the same capacity of migration. Subsequently, the improvement of oral hygiene and oral health should be considered as protective factors against the risk of bacteremia due to this specific pathogen, especially in patients at risk. In fact, in the presence of active periodontal pockets, bacteremia can be determined by routine daily activities such as tooth brushing, with a major cumulative exposition rate that poses a similar risk to dental invasive procedures [18]. Moreover, the bacterial invasion from periodontal tissues to endodontic tissues [19] can determine another dissemination source of bacteria through pulpal and periapical blood vessels into systemic bloodstream. The present study has some limitations, especially regarding the number of samples, as the need to analyze only samples without caries has conditioned the number of samples. Therefore, further studies will be needed to expand the sample and evaluate possible variables such as age, geographical origin, etc.

## 5. Conclusions

The *S. mutans* is considered an important oral pathogen and it is also recognized to have the ability to invade via bloodstream different tissues and organs, thus promoting systemic diseases. Therefore, their precise investigation using advanced techniques is strongly recommended in patients affected by EPL, and the use of advanced techniques could also be useful to investigate the possible presence of CBPs genes, which could enhance the virulence of specific isolates.

**Author Contributions:** Conceptualization, A.B. and A.T.; Methodology, A.T., M.C. (Marco Cordisco) and M.M.D.; Formal analysis and investigation, M.C. (Michele Camero) and M.M.D.; Writing—original draft preparation, A.T., A.B. and M.C. (Marco Cordisco); Writing—review and editing, M.C. (Michele Camero) and M.C. (Marialaura Corrente); Resources, M.C. (Michele Camero) and M.C. (Marialaura Corrente); Supervision, M.C. (Marialaura Corrente). All authors have read and agreed to the published version of the manuscript.

**Funding:** This research received no external funding.

**Institutional Review Board Statement:** The patients gave informed consent for inclusion before they participated in the study. The study was conducted in accordance with the Declaration of Helsinki of 1975, revised in 2013. The protocol did not require approval from Ethics Committee, because the samples were taken during the clinical routinely protocol.

**Informed Consent Statement:** Written informed consent has been obtained from the patients.

**Data Availability Statement:** The data presented in this study are available in Section 2.

**Conflicts of Interest:** The authors declare no conflict of interest.

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
