# Peer review of "Streptococcus mutans Associated with Endo-Periodontal Lesions in Intact Teeth"

_applsci, doi:10.3390/app122211837_

Round 1

Reviewer 1 Report

The content is interesting; however, minor adjustments are needed.

-          The title doesn’t clearly reflect the content of the paper.

-          Abstract:

 Line16: The word  (thanks ) needs to be changed. It doesn’t match the negative role of collagen.

There are a few abbreviations used without explanation: for example qPCR

-          Introduction

Line 31: ( harboring of about 1010 microorganisms.) A reference is needed.

Line 42: Caries is a result of (oral care and hygiene,) or of (bad oral care and hygiene)?

Results

Adding graphics to the study findings can improve the reader's comprehension.

Discussion

Line 257-258: The sentence may cause misleading to the reader and need rephrasing. The authors rely on their study (which has a very small sample size) to say that one technique is more sensitive than another.

Line 267-269: The sentence needs to be deleted.

The study result was, (S. mutans involved in periodontal disease can vehicle specific genes that could enhance its virulence and capacity of systemically disseminate in other organs) without clear statistical evidence.

Author Response

To the Editor of

Applied Science

Dear Editor

I send you the revised manuscript:Streptococcus mutans associated with endo-periodontal lesions in intact teeth

by Buonavoglia et al.

We have found the Reviewers’ comments very helpful and we believe we have addressed each question made by the Reviewers in the revised manuscript enclosed here. Please find below a point-by-point reply to each reviewer.

Best Regards,

Dr. Adriana Trotta

Valenzano (BA), 18/11/2022

Dr. Adriana Trotta, DVM, Post-doc researcher, Department of Veterinary Medicine, University of Bari “Aldo Moro”, Str. Prov. per Casamassima, Km 3, 70010 Valenzano, BA, Italy Email contacts: adriana.trotta@uniba.it; trotta.adri@gmail.com

Answers to Reviewers’ comments.

Reply to Reviewer n. 1 comments

Q1: Comments and Suggestions for Authors

The content is interesting; however, minor adjustments are needed.

A1: Thank you for you revision and comment

Q2: The title doesn’t clearly reflect the content of the paper.

A2: Accordingly, we have modified the Title. (Please, see lines 2-3).

-Abstract:

Q3: Line16: The word (thanks) needs to be changed. It doesn’t match the negative role of collagen.

A3: Accordingly, the word has been changed (Please, see line 16).

Q4: There are a few abbreviations used without explanation: for example, qPCR

A4: Accordingly, the abbreviations have been explained (Please, see lines 20 and 21).

-Introduction:

Q5: Line 31: (harboring of about 1010 microorganisms.) A reference is needed.

A5: Thank you for your comment. We added the reference (Please, see line 31).

Q6: Line 42: Caries is a result of (oral care and hygiene,) or of (bad oral care and hygiene)?

A6: Thank you for your suggestion. The phrase has been changed (Please, see line 41).

-Results:

Q7: Adding graphics to the study findings can improve the reader's comprehension.

A7: Thank you for your comment that we find very relevant, moreover we think that adding the graphs would be a repetition of the results reported in the table, and we deem it more appropriate to group the data in a single explanatory table rather than split them into different graphs.

-Discussion:

Q8: Line 257-258: The sentence may cause misleading to the reader and need rephrasing. The authors rely on their study (which has a very small sample size) to say that one technique is more sensitive than another.

A8: Thank you for the comment which we find very helpful. Accordingly, the text has been improved (Please, see lines 268-272).

Q9: Line 267-269: The sentence needs to be deleted.

The study result was (S. mutans involved in periodontal disease can vehicle specific genes that could enhance its virulence and capacity of systemically disseminate in other organs) without clear statistical evidence.

A9: Accordingly, the phrase has been deleted (Please, see line 282-285).

Reviewer 2 Report

Methods

Ethical waiver should be obtained from the Human Ethics Comittee.

Line 87 says'The exclusion criteria were the presence of caries' while Figure 1 shows a carious lesion of the cementum.

Line 117 state the transport media.

Line 131- 134- Sentence too long and looses substance, break by putting full stop after Italy and start with 'The swabs were collected............

line 133 - add paper point.

line 135-136 CBA non selective media, MCK gram -ve bacilli, MSA Staphylococci id., TSB non-selective fastidious: why all these media? S. mutans can be identified with Mutans Bacitrasin Agar.

Line 140- which Api Strept systems?

143: kindly provide subheadings on the experminets perfomed i.e S. mutans Reverse transcriptase polymerase chain reaction.

178 provide the name of the lab strain used.

Line 143- 147 Sentence too long and therefore break it down.

144 replace using with and full stop after supplier.

146 and co workers et al., (year).

Results

Give subheadings as well.

line 194-195 (34) is like a reference, therefore you can say'among the 34 samples tested'.

194-197 sentence too long

line 205 - remove 'S. mutans positive sentence......' its the repeted on the next sentence.

discussion 

243- negative caries teeth, please fix as stated in methodoloy.

264- which method are you talking about for isolation of S. mutans in 50% of samples?

Author Response

Reply to Reviewer n. 2 comments

- Methods

Q1: Ethical waiver should be obtained from the Human Ethics Comittee.

A1: Thank you for your revision and comment, but the protocol did not require approval from Ethics Committee, because the samples were taken during the clinical routinely protocol.

Q2: Line 87 says 'The exclusion criteria were the presence of caries' while Figure 1 shows a carious lesion of the cementum.

A2: Thank you for your necessary revision. We agree with the referee as there was an error to adding a correct picture in the text. We changed the picture in Figure 1 and this is correct (Please see line 11-112).

Q3: Line 117 state the transport media.

A3: Accordingly, we have added the transport medium (Please, see line 121).

Q4: Line 131- 134- Sentence too long and looses substance, break by putting full stop after Italy and start with 'The swabs were collected............

A4: Thank you for the comment. Accordingly, the phrase has been modified (Please, see line 138).

Q5: line 133 - add paper point.

A5: Accordingly, the word has been added (Please, see line 139).

Q6: line 135-136 CBA non selective media, MCK gram -ve bacilli, MSA Staphylococci id., TSB non-selective fastidious: why all these media? S. mutans can be identified with Mutans Bacitrasin Agar.

A7: Thank you for the comment which we find very relevant. In the study we chose a culture-method protocol that could, if there were any, let to detect the presence of other pathogenic bacterial species, besides the S. mutans, whose presence could have influenced the results of the study. For this reason, we have clarified the method section (Please, see line 140-141).

Q7: Line 140- which Api Strept systems?

A7: Accordingly, the phrase has been modified (Please, see line 147).

Q8: 143: kindly provide subheadings on the experiments performed i.e S. mutans Reverse transcriptase polymerase chain reaction.

A8: Thank you for the suggestion. Accordingly, the paragraph has been divided. (Please, see line 149 and 167).

Q9: 178 provide the name of the lab strain used.

A9: Thank you for your comment. The strains used as controls in this case came from laboratory collection, therefore they are identified with an internal protocol number, which we believe does not add any useful information for the purposes of the study, as it is referred to an internal laboratory database.

Q10: Line 143- 147 Sentence too long and therefore break it down.

A10: Accordingly, the phrase has been modified (Please, see line 151).

Q11: 144 replace using with and full stop after supplier.

A11: Accordingly, the phrase has been modified (Please, see line 151-154).

Q12: 146 and co-workers et al., (year).

A12: Accordingly, the year has been added (Please, see line 154).

-Results

Q13: Give subheadings as well

A13: Accordingly, we have added the subheading (Please, see line 213).

Q14: line 194-195 (34) is like a reference, therefore you can say'among the 34 samples tested'.

A14: Thank you for your comment. The phrase has been changed (Please, see line 203).

Q15:194-197 sentence too long

A15: Thank you for your comment. The phrase has been changed (Please, see lines 203).

Q16: line 205 - remove 'S. mutans positive sentence......' its the repeted on the next sentence.

A16: Accordingly, the phrase has been changed (Please, see line 214).

-Discussion

Q17: 243- negative caries teeth, please fix as stated in methodoloy.

A17: Thank you for the comment. Accordingly, the phrase has been modified (Please, see lines 248-253).

Q18: 264- which method are you talking about for isolation of S. mutans in 50% of samples?

A18: Thank you for the comment. We were talking about the culture-based method specified in the method section, but the phrase has been modified to make it clearer (Please, see lines 278-282).

Reviewer 3 Report

Dear Editors of the Applied Sciences Journal

I trust you are well.

Kindly find enclosed my recommendations for the submitted manuscript entitled “Collagen binding proteins of Streptococcus mutans associated with endo-periodontal lesions in intact teeth”.

Many thanks for the opportunity to review the research study on behalf of the Applied Sciences Journal.

Kind regards

Dr. Musa Marimani

Author Response

Reply to Reviewer n. 3 comments

Comments:

The reviewed manuscript is entitled: “Collagen binding proteins of Streptococcus mutans

associated with endo-periodontal lesions in intact teeth”.

The manuscript is easy to follow and comprehend, very descriptive, informative and is written in appropriate scientific and clinical language.

My comments are as follows:

Specific comments for the submitted paper:

-Method section:

2.1. Patient`s information

Q1: (a) Please motivate why a sample size of 17 (n=17) was used in the study.

A1: Thank you for the comment. The present study has limitations especially regarding the number of samples, since the need to analyze only patients without caries has influenced the number of samples. Accordingly, we have stated this at the lines 318-321.

Q2: (b) Figure 1 in line 108 and line 110-112 needs to be written in a more appropriate scientific

language.

A2: Thank you for the comment.  We have modified the phrase (Please, see lines 113).

-Section 3: Results:

Q3: (a) Since qPCR and conventional PCR are explained in the “materials and methods” section (section 2.4), it would be very useful to show the qPCR data graph or the agarose gel

electrophoresis image in the “results” section (section 3).

A3: Thank you for the suggestion which we find very useful, but we think that graphs of tests in this case would not add any additional information to the work, respecting to what has already been described in material and results section.

Q4: (b) Statistical analysis for the qPCR data needs to be clearly outlined in “methods” and “results” sections.

A4: Thank you for the comment. Accordingly, we added this data in lines 211-212.

-Section 4: Discussion:

Q5: (a) Why was the gtfB gene specifically used for detection of S. mutans DNA by PCR? Please

provide a specific purpose for this in the “discussion” section (section 4).

A5: Thank you for the comment. Accordingly, the discussion section has been improved (Please, see lines 268).

Q6: (b) The function of the Cnm gene is mentioned in the “discussion” section. Provide the function of the Cbm gene in S. mutans in the “discussion” section (section 4).

A6: Thank you for your comment. Accordingly, the manuscript has been improved (Please see lines 300-302).

Q7: (c) The authors need to articulate the limitations of the study in the “discussion section” (section 4) such as sample size, age, demographics of participants etc. 2

A7: Thank you for your comment. Accordingly, the text has been improved (Please, see lines 318-321).

Q8: (d) The authors need to state how the study limitations will be addressed in prospective research investigations.

A8: Thank you for your comment. Accordingly, the text has been improved (Please, see lines 318-321).

Reviewer 4 Report

The article is interesting and well written.

1. Provide reference for S. mutans strains are classified into… line 54 – 60.

2. Rewrite conclusions based on objective of the study.

3. Is there a difference in numbers of S. mutans in patients with and without pockets, in general – literature evidence

4. Can there be similarity in S. intermedius and S. mutans and possibility of false positives with PCR primers

5. Line 163: d cbm-Reverve (5’-….    Correct spelling

6. Line 185: 3 µl of sterile RNAasi and DNAasi free water)…. Check for correctness

7. Compare observations from culture method with PCR and sequencing.

8. Reasons for non detection of cbm gene and very few sites with cnm. Are these relevant to adhesion to hard tissue.

9. Can necrosis of pulp within the root canal explain low presence of these genes required for microbial adhesion.

10. Does the presence of an acidic environment affect the expression of certain genes.

11. Will root canal irrigation and subgingival irrigation have an impact on treatment of advanced perio-endo lesions.

12. Clearly state objectives of study

13. No sample size calculation has also been done.

14. The study involved 17 patients from whom extracted tooth samples were obtained. No ethical approval was obtained for carrying out the study. This may be required. Is this part of a larger research work. Or Is this a case series.

Author Response

Reviewer n. 4

The article is interesting and well written.

Thank you for your revision and comment which we find very helpful.

Q1: Provide reference for S. mutans strains are classified into… line 54 – 60.

A1: Thank you for your revision and comment. Accordingly, we have added the reference number at the line 58.

Q2. Rewrite conclusions based on objective of the study.

A2: Thank you for your comment. Accordingly, the text has been improved (Please, see lines 323-328).

Q3: Is there a difference in numbers of S. mutans in patients with and without pockets, in general – literature evidence

A3: Thanks for your comment. Previous studies (Iwano Y, Sugano N, Matsumoto K, Nishihara R, Iizuka T, Yoshinuma N, Ito K. Salivary microbial levels in relation to periodontal status and caries development. J Periodontal Res. 2010 Apr;45(2):165-9.) demonstrated that salivary levels of S. mutans are significantly higher in the periodontally healthy patients than in patients affected by periodontal disease. Moreover, other studies (Van der Reijden WA, Dellemijn-Kippuw N, Stijne-van Nes AM, de Soet JJ, van Winkelhoff AJ. Mutans streptococci in subgingival plaque of treated and untreated patients with periodontitis. J Clin Periodontol. 2001 Jul;28(7):686-91.) demonstrated that in patients affected by periodontal disease, the prevalence of S. Mutans in subgingival plaque was higher in untreated patients with active disease respect to treated patients in maintenance phase.

Q4: Can there be similarity in S. intermedius and S. mutans and possibility of false positives with PCR primers

A4: Thank you for the comment but, it is not possible, because these primer set is specie-specific for S. mutans because targeting the highly conserved region of gene gtfB as stated in the references used for the test and in the manuscript at lines 269-270, which were improved to clarify.

Q5: Line 163: d cbm-Reverve (5’-….    Correct spelling

A5: Corrected (Please, see line 171).

Q6: Line 185: 3 µl of sterile RNAasi and DNAasi free water)…. Check for correctness

A6: Corrected (Please, see line 193).

Q7: Compare observations from culture method with PCR and sequencing.

A7: Thank you for the comment. You can find this information at the lines 268-272, in which, accordingly to your revision, the text has been improved.

Q8. Reasons for non detection of cbm gene and very few sites with cnm. Are these relevant to adhesion to hard tissue.

A8: Thank you for the comment, moreover the question is still controversial. In our opinion, the absence of cbm-positive strains, is not surprising since some different studies reported the cbm+ strains as 2% or 3% of the bacterial population, compared to 15-20% of Cnm+ strains (Nomura, R.; Nakano, K.; Naka, S.; Nemoto, H.; Masuda, K.; Lapirattanakul, J.; Alaluusua, S.; Matsumoto, M.; Kawabata, S.; Ooshima, T. Identification and characterization of a collagen-binding protein, Cbm, in Streptococcus mutans. Mol. Oral Microbiol 2021, 27(4):308-23. doi: 10.1111/j.2041-1014.2012.00649.x. and Otsugu, M., Mikasa, Y., Kitamura, T. et al. Clinical characteristics of children and guardians possessing CBP-positive Streptococcus mutans strains: a cross-sectional study. Sci Rep 12, 17510 (2022). https://doi.org/10.1038/s41598-022-22378-8). In this study, the limited number of enrolled patients may have influenced this result and therefore we felt it appropriate to indicate the study limits at the lines 318-321.

Q9. Can necrosis of pulp within the root canal explain low presence of these genes required for microbial adhesion.

A9: Thank you for the comment which we find very helpful, moreover the question is still unclear and, in this study, we hypothesize that the low detection of the genes could be due to the limited number of samples (which is a limit of the study as stated at lines 318-321) rather than other factors such as necrosis. Moreover, we hypothesize that some acquired properties obtained from this gene (adherence to endothelial and epithelial cells, intracellular invasion) have less utility in necrotic endodontic tissues (Please, see lines 298-299).

Q10. Does the presence of an acidic environment affect the expression of certain genes.

A10: Thank you for your comment. The environmental factors that can affect the presence and the expression of cnm and cbm genes are still largely unknown and in a recent study (Otsugu, M., Mikasa, Y., Kitamura, T. et al. Clinical characteristics of children and guardians possessing CBP-positive Streptococcus mutans strains: a cross-sectional study. Sci Rep 12, 17510 (2022). https://doi.org/10.1038/s41598-022-22378-8) the authors have started to investigate some clinical characteristics of patients with S. mutans CBP-positive strains but the acidity of the oral environment it was not considered as a predisposing factor.

Q11. Will root canal irrigation and subgingival irrigation have an impact on treatment of advanced perio-endo lesions.

A11: Thanks for the relevant comment. Root canal disinfection with irrigants have a general evidenced effect on bacteria in root canal system with effectiveness on endodontic component prognosis of endo-perio lesion. Subgingival irrigation should be associated with a mechanical removal of root surface-associated biofilms for an effective impact on treatment of periodontal component of perio-endo lesions.

Q12. Clearly state objectives of study

A12: Thank you for the comment. Accordingly, the objectives of the study have been clearly stated at the lines 78-83.

Q13. No sample size calculation has also been done.

A13: The calculation of the sample size was not done as one of the limitations of the study was precisely the related sample size (Please, see lines 318-321).

Q14. The study involved 17 patients from whom extracted tooth samples were obtained. No ethical approval was obtained for carrying out the study. This may be required. Is this part of a larger research work. Or Is this a case series.

A14: Thank you for the observation. This is a preliminary study which needs further investigation in order to develop a larger research project. The protocol did not require approval from Ethics Committee, because the samples were taken during the clinical routinely protocol.
